DATA RELEASE

# Genomic features of *Mycobacterium avium* subsp. *hominissuis* isolated from pigs in Japan

Tetsuya Komatsu[1,†], Kenji Ohya[2,3,†,‡], Atsushi Ota[4], Yukiko Nishiuchi[5], Hirokazu Yano[6], Kayoko Matsuo[7], Justice Opare Odoi[3], Shota Suganuma[2,§], Kotaro Sawai[2,8], Akemi Hasebe[9], Tetsuo Asai[3], Tokuma Yanai[2,3,10], Hideto Fukushi[2,3], Takayuki Wada[11], Shiomi Yoshida[12], Toshihiro Ito[13], Kentaro Arikawa[14], Mikihiko Kawai[15], Manabu Ato[16], Anthony D Baughn[17], Tomotada Iwamoto[14] and Fumito Maruyama[5,18,19,*]

1 Aichi Prefectural Chuo Livestock Hygiene Service Center, Okazaki, Aichi, Japan
2 Faculty of Applied Biological Sciences, Gifu University, Gifu, Gifu, Japan
3 United Graduate School of Veterinary Sciences, Gifu University, Gifu, Gifu, Japan
4 Data Science Center, Division of Biological Science, Nara Institute of Science and Technology, Ikoma, Nara, Japan
5 Office of Academic Research and Industry-Government Collaboration, Hiroshima University, Higashi-Hiroshima, Hiroshima, Japan
6 Graduate School of Life Sciences, Tohoku University, Sendai, Miyagi, Japan
7 Kumamoto Prefectural Aso Public Health Center, Aso, Kumamoto, Japan
8 Viral Disease and Epidemiology Research Division, National Institute of Animal Health, National Agriculture Research Organization, Tsukuba, Ibaraki, Japan
9 Toyama Prefectural Meat Inspection Center, Imizu, Toyama, Japan
10 Hiwa Natural History Museum, Shobara, Hiroshima, Japan
11 Graduate School of Human Life Science, Osaka City University, Osaka, Osaka, Japan
12 Clinical Research Center, National Hospital Organization Kinki-Chuo Chest Medical Center, Sakai, Osaka, Japan
13 Laboratory of Proteome Research, Proteome Research Center, National Institutes of Biomedical Innovation, Health and Nutrition, Ibaraki, Osaka, Japan
14 Department of Infectious Diseases, Kobe Institute of Health, Kobe, Hyogo, Japan
15 Graduate School of Human and Environmental Studies, Kyoto University, Kyoto, Kyoto, Japan
16 Department of Mycobacteriology, Leprosy Research Center, National Institute of Infectious Diseases, Higashimurayama, Tokyo, Japan
17 Department of Microbiology and Immunology, University of Minnesota Medical School, Minneapolis, Minnesota, USA
18 Project Research Center for Holobiome and Built Environment (CHOBE), Hiroshima University, Higashi-Hiroshima, Hiroshima, Japan
19 Scientific and Technological Bioresource Nucleus, Universidad de La Frontera, Temuco, Chile

**Submitted:** 08 June 2021

* Corresponding author. E-mail: fumito@hiroshima-u.ac.jp

† Contributed equally.

‡ Current address: National Institute of Health Sciences, Kawasaki, Kanagawa, Japan.

§ Current address: Central Research Institute for Feed and Livestock of Zen-noh, Tsukuba, Ibaraki, Japan.

Preprint submitted at https://doi.org/10.1101/2021.06.15.447579

## ABSTRACT

*Mycobacterium avium* subsp. *hominissuis* (MAH) is one of the most important agents causing non-tuberculosis mycobacterial infection in humans and pigs. There have been advances in genome analysis of MAH from human isolates, but studies of isolates from pigs are limited despite its potential source of infection to human. Here, we obtained 30 draft genome sequences of MAH from pigs reared in Japan. The 30 draft genomes were 4,848,678–5,620,788 bp in length, comprising 4652–5388 coding genes and 46–75 (median: 47) tRNAs. All isolates had restriction modification-associated genes and 185–222 predicted virulence genes. Two isolates had tRNA arrays and one isolate had a clustered regularly interspaced short palindromic repeat (CRISPR)

region. Our results will be useful for evaluation of the ecology of MAH by providing a foundation for genome-based epidemiological studies.

**Subjects** Microbiology, Molecular Genetics, Microbial Ecology

## DATA DESCRIPTION

### Context

To date, incidence of infection caused by non-tuberculous mycobacteria (NTM) has been increasing worldwide [1]. Among NTMs, *Mycobacterium avium* complex (MAC) is one of the most critical agents. *M. avium* has four subspecies: *M. avium* subsp. *avium* (MAA), *M. avium* subsp. *paratuberculosis* (MAP), *M. avium* subsp. *silvaticum* (MAS) and *M. avium* subsp. *hominissuis* (MAH). MAH is a major pathogen for humans, causing lung disease and sometimes disseminated infection in immune-suppressed patients [2, 3]. MAH is also a main causative agent of mycobacteriosis in pigs [4], showing mesenteric and mandibular lymphadenitis [5] and sometimes systemic infection [6]. Swine mycobacteriosis has severe economic effects on affected farms. MAH-infected pigs are suspected to be a potential risk for human infection [7–10].

Recently, there has been extensive progression in the genomic epidemiological study of MAH. Based on findings from our recent studies, MAH is divided into six major lineages: MahEastAsia1, MahEastAsia2, and SC1–4. Each lineage is predominant in specific regions on a global scale [11, 12]. For example, the MahEastAsia1 and MahEastAsia2 lineages are frequently isolated from human lung disease in Japan and Korea, but SC1–4 lineages are isolated from America and Europe [11, 12]. Japanese pig isolates are mainly classified into two lineages, SC2 and SC4 [11, 12]. However, the number of pig isolates used in these studies was insufficient to precisely clarify the ecology of MAH.

Most of the essential genes of MAH are thought to be mutual orthologs of genes in *Mycobacterium tuberculosis* (MTB) [13]. Although components of virulence systems have been investigated [14], reports about genome contents, even drug resistance genes are not available, despite the increasing incidence of MAH disease [1]. To understand MAH evolution and distribution, and to promote the identification of targets for antimicrobial drug discovery, characterization of the defining genomic features of MAH is essential.

Here, we obtained draft genome sequences of 30 MAH (NCBI:txid439334) isolates derived from pigs reared in Japan, and identified genome features for bacterial defense systems, such as restriction modification (RM) system, clustered regularly interspaced short palindromic repeat (CRISPR), tRNA arrays, virulence factors and drug resistance genes. The results from this study may enable greater understanding of the epidemiological relationship between MAH in humans and pigs.

### Methods

Protocols for bacterial isolation and DNA extraction are available in a protocols.io collection (Figure 1 [15]).

#### *Sampling*

MAH isolates were collected from pigs reared at two areas, Tokai and Hokuriku in Japan, where about 10% of pigs in Japan are reared. Forty-eight mesenteric or mandibular lymph

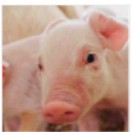

### Bacterial isolation and genomic DNA extraction of Mycobacterium avium from pig lymph nodes ▾

Tetsuya Komatsu[1], Kenji Ohya[2], Justice Opare Odoi[3], Shota Suganuma[2], Kotaro Sawai[2], Takayuki Wada[4], Tomotada Iwamoto[5], Fumito Maruyama[6]

[1]Aichi Prefectural Chuo Livestock Hygiene Service Center, Okazaki, Aichi, Japan;

[2]Faculty of Applied Biological Sciences, Gifu University, Gifu, Gifu, Japan;

[3]United Graduate School of Veterinary Sciences, Gifu University, Gifu, Gifu, Japan;

[4]Graduate School of Human Life Science, Osaka City University, Osaka, Osaka, Japan;

[5]Department of Infectious Diseases, Kobe Institute of Health, Kobe, Hyogo, Japan;

[6]Office of Academic Research and Industry-Government Collaboration, Hiroshima University, Higashi-Hiroshima, Hiroshima, Japan

dx.doi.org/10.17504/protocols.io.bzbtp2nn

Tetsuya Komatsu
Aichi Prefectural Chuo Livestock Hygiene Service Center, Oka...

**Figure 1.** Protocols for bacterial isolation and genomic DNA extraction of *Mycobacterium avium* from pig lymph nodes [15]. https://www.protocols.io/widgets/doi?uri=dx.doi.org/10.17504/protocols.io.bzbtp2nn

nodes of pigs reared in the Tokai area were collected from Gifu Meat Inspection Center between July and December 2015. Samples (20 mesenteric lymph nodes, one mandibular lymph node, one liver) from Tokai and Hokuriku areas were collected between August 1998 and March 2018 and archived at Toyama Meat Inspection Center.

### Bacterial isolation and DNA extraction

The method used for bacterial isolation is available in protocols.io [16]. Mesenteric or mandibular lymph nodes with mycobacterial granulomatous lesions were mixed with 400 µl of 2% NaOH and incubated at room temperature overnight. The samples were spread onto 2% Ogawa medium (Kyokuto Pharmaceutical, Tokyo, Japan) and incubated at 37 °C for 3–4 weeks. A single colony was inoculated onto 7H11 broth with 10% oleic acid-albumin-dextrose-catalase as a supplement. The isolates were stored with Microbank (Pro Lab Diagnostics Inc., Richmond Hill, ON, Canada) at –80 °C. The method of extraction of genomic DNA was also available in protocols.io [17]. In brief, cells were delipidated by treatment with acetone, then lysed by lysozyme and Proteinase K. Genomic DNA was extracted by phenol/chloroform treatment of the lysates.

### Identification of MAH and insertion sequence profile

PCR amplification of *M. avium* 16S rRNA genes (MAV) was conducted for screening [18]. Isolates positive for MAV were identified by sequencing *hsp65* and *rpoB* genes [19, 20]. Basic Local Alignment Search Tool (BLAST) analysis was conducted using partial sequences of *rpoB* gene. Phylogenetic analysis of both genes was conducted using the maximum likelihood method using Molecular Evolutionary Genetics Analysis (MEGA) software v.7.0 (RRID:SCR_000667). Bootstrap values were calculated from 1000 replications. Insertion sequence patterns of IS*900*, IS*901*, IS*902* and IS*1245* were performed as described previously [21–23]. IS*1311* and IS*1613* were searched for within draft genomes using ISfinder v.2.0 (RRID:SCR_003020) [24] with default parameters [25].

### Draft genome sequences and genome annotation

Extraction of genomic DNA was described above. An average of 350-bp paired-end libraries were prepared from extracted genomic DNA using the TruSeq DNA PCR-Free High

Throughput Library Prep Kit (Illumina, San Diego, CA, USA). Paired-end sequencing (2 × 150 bp) was conducted using the HiSeq X Ten sequencing platform (Illumina) at the Beijing Genomics Institute (Shenzhen, China). Output reads were trimmed by TrimGalore v.0.4.1 (RRID:SCR_011847) [26] and mismatched reads were corrected by SPAdes v.3.12.0 (RRID:SCR_000131) [27]. The reads were assembled and polished using Pilon v.1.22 (RRID:SCR_014731) [28] and Unicycler v.0.4.6 [29], and genome completeness was then estimated by CheckM v.1.0.7 (RRID:SCR_016646) [30]. Taxonomic classification of contigs was carried out using Kaiju v1.4.5 [31] and Anvi'o v.3 [32]. Draft genome sequences were annotated via the National Center for Biotechnology Information (NCBI) Prokaryotic Genome Annotation Pipeline (PGAP) v.4.9 (RRID:SCR_021329) [33].

### Detection of bacterial defence systems (RM system and CRISPR CAS system) in the MAH genome

RM systems were determined using the online tool Restriction-ModificationFinder v.1.1 [34] twice, with the following settings (1: database: All incl. putative genes, threshold for %ID: 90%, minimum length: 80% to search the RM system of MAH and 2: database: All, threshold for %ID: 10%, minimum length: 20% to confirm the orthologue of MTB or the other Mycobacteria) [35]. CRISPR-Cas9 systems were identified using the online tool CRISPRCasFinder v.4.2.2 [36] with default setting [37, 38].

### Detection of tRNA arrays in the MAH genome

The total number of tRNAs in this study were retrieved from gb files annotated by PGAP. Draft genomes of GM17 and OCU479 isolates, which had more tRNAs than the others (Table 1), were inspected by tRNAscan-SE v.2.0 (RRID:SCR_010835) to search tRNA arrays [39]. tRNA gene isotype synteny (expressed by the single-letter amino acid code) of both isolates and the reference strains were aligned and used for the maximum likelihood method by MEGA 7.0. Classification of both isolates was conducted as previously described [40].

### Detection of virulence factors and drug resistance genes

Virulence genes were identified using VFanalyzer (release 5) [41, 42]. We selected the following settings: genus: Mycobacterium, specify a representative genome: *M. avium* 104 and choose genomes for comparison: blank. Draft genome fasta files were uploaded. Drug resistance genes were identified by Resistance Gene Identifier (RGI) v.5.1.0 [43] with the following settings: Select Data Type: DNA sequence, Select Criteria: Perfect and Strict hit only, Nudge ≥95% identity Loose hits to Strict: Exclude nudge, Sequence Quality: high quality/coverage [44]. To confirm the existence of mutations detected by RGI, we retrieved the respective drug resistance-associated genes from draft genome sequences, aligned by MEGA 7.0, and then manually checked for mutations in the nucleotide sequences.

## DATA VALIDATION AND QUALITY CONTROL

### Identification of MAH

The experimental workflow from sampling to identification is shown in Figure 2. We successfully obtained 13 MAH isolates derived from the Tokai area. Of these, 8 isolates

**Table 1.** Summary information for the draft genome sequences of 30 MAH isolates in this study.

| Isolate | Genome size (bp) | N50 (bp) | Coverage | No. of contigs | G+C content (%) | No. of CDSs* | No. of tRNAs |
|---|---|---|---|---|---|---|---|
| GM5 | 5,037,010 | 35,760 | 277 | 224 | 69.06 | 4,877 | 47 |
| GM10 | 4,858,055 | 33,212 | 277 | 248 | 69.16 | 4,708 | 47 |
| GM12 | 4,848,678 | 33,219 | 253 | 261 | 69.17 | 4,732 | 47 |
| GM16 | 5,012,047 | 24,262 | 274 | 346 | 68.84 | 4,981 | 46 |
| GM17 | 5,265,075 | 30,906 | 355 | 289 | 68.77 | 5,190 | 75 |
| GM21 | 4,899,737 | 45,080 | 411 | 216 | 69.20 | 4,734 | 47 |
| GM32 | 4,897,271 | 47,147 | 292 | 208 | 69.20 | 4,712 | 47 |
| GM44 | 5,086,547 | 26,307 | 251 | 316 | 68.95 | 4,780 | 46 |
| OCU467 | 5,110,693 | 243,182 | 207 | 75 | 69.16 | 4,803 | 46 |
| OCU468 | 5,459,638 | 137,464 | 198 | 132 | 68.96 | 5,176 | 46 |
| OCU469 | 5,167,480 | 190,329 | 191 | 57 | 69.19 | 4,886 | 47 |
| OCU470 | 5,388,572 | 124,661 | 220 | 132 | 68.98 | 5,103 | 46 |
| OCU471 | 4,990,913 | 193,095 | 237 | 70 | 69.24 | 4,713 | 47 |
| OCU472 | 5,410,552 | 119,264 | 180 | 139 | 68.97 | 5,163 | 47 |
| OCU473 | 5,237,229 | 105,027 | 232 | 118 | 69.11 | 4,981 | 47 |
| OCU474 | 5,087,878 | 168,670 | 213 | 81 | 69.26 | 4,817 | 47 |
| OCU475 | 5,376,580 | 113,114 | 243 | 130 | 68.99 | 5,121 | 46 |
| OCU476 | 5,359,545 | 133,302 | 268 | 132 | 69.00 | 5,094 | 46 |
| OCU477 | 5,087,664 | 218,065 | 221 | 85 | 69.22 | 4,779 | 47 |
| OCU478 | 5,108,303 | 272,265 | 230 | 73 | 69.17 | 4,803 | 46 |
| OCU479 | 5,620,788 | 112,152 | 167 | 143 | 68.78 | 5,388 | 75 |
| OCU480 | 5,088,946 | 195,446 | 53 | 73 | 69.24 | 4,820 | 47 |
| OCU481 | 5,100,722 | 163,519 | 247 | 101 | 69.19 | 4,802 | 47 |
| OCU482 | 5,100,769 | 163,705 | 244 | 99 | 69.19 | 4,800 | 47 |
| OCU483 | 4,943,024 | 200,611 | 228 | 68 | 69.24 | 4,652 | 47 |
| OCU484 | 5,096,430 | 141,792 | 249 | 104 | 69.20 | 4,811 | 47 |
| OCU485 | 5,109,020 | 243,182 | 258 | 80 | 69.16 | 4,805 | 46 |
| OCU486 | 5,023,805 | 234,302 | 40 | 52 | 69.23 | 4,722 | 47 |
| Toy194 | 5,347,524 | 216,164 | 273 | 93 | 68.97 | 5,018 | 47 |
| Toy195 | 5,346,468 | 168,809 | 192 | 103 | 68.97 | 5,029 | 47 |

*CDSs: coding sequences.

(GM5–GM44), together with 22 isolates from Tokai and Hokuriku areas (OCU467–OCU486, Toy194 and Toy195) were used for draft genome sequence analysis. We conducted multiple examinations to determine the isolates as MAH, IS possession patterns, or sequence analysis of *hsp65* [45]. Among MAH subspecies, the patterns of IS possession are different and are used for subspecies identification [46]. IS*900* and IS*901* are known to be indicators of MAP and MAA, respectively [22, 23]. MAH is usually positive for IS*1245* [47] and is negative for IS*900*, IS*901* and IS*902* [21]; however, MAH strains without IS*1245* are frequently distributed in Japan [46, 48]. In our study, 10/30 isolates were negative for IS*1245* (33.3%) and none had IS*900*, IS*901* and IS*902* [45]. Subspecies of *M. avium* are also usually identified by *hsp65* gene analysis, which had 17 single nucleotide polymorphisms (SNP) variations among subspecies [20]. MAH usually has 1, 2, 3, 7, 8 or 9 *hsp* codes [20]; however, five isolates had unclassified *hsp* codes (indicated by N) in this study [45]. Therefore, we also conducted partial sequence analysis of the *rpoB* gene and the isolates were identified as MAH by BLAST analysis. In addition, we conducted phylogenetic analysis based on *hsp65* and *rpoB* genes retrieved from the draft genome, and all isolates in this study were also classified as MAH (Figure 3). All these examinations confirmed that our isolates were MAH.

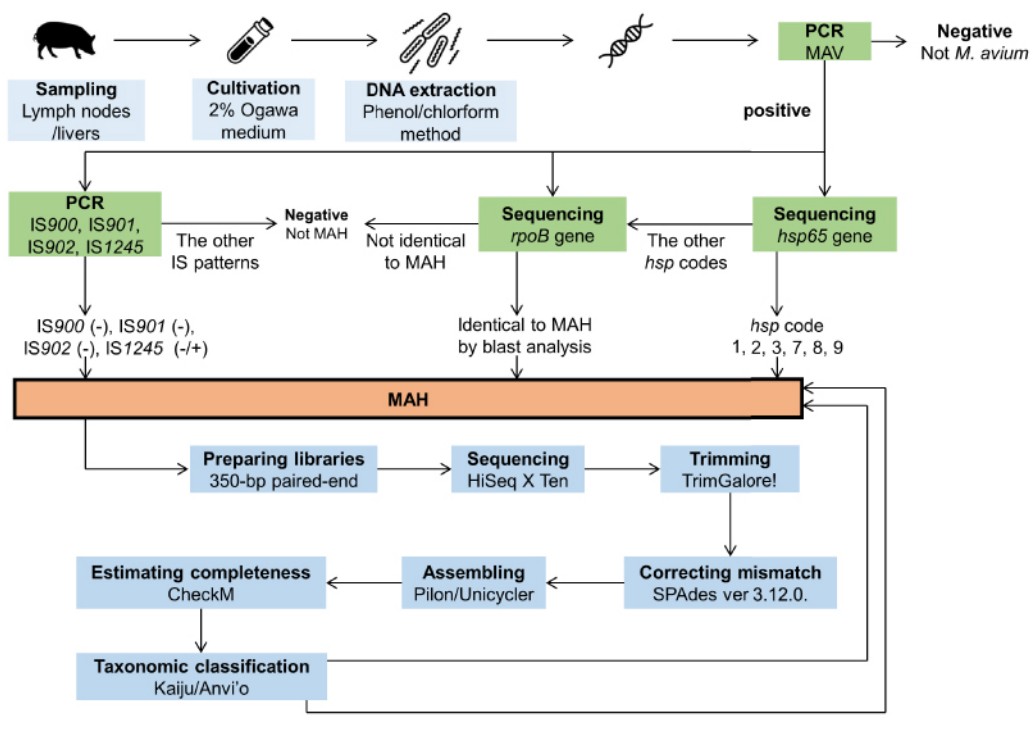

**Figure 2.** The experimental workflows in this study.

## Draft genome data

All our draft genome sequences had a total length of between 4.85 and 5.62 Mbp (megabase pairs), similar to complete MAH genomes [49, 50]. All isolates had N50 values over 24 Kbp (kilobase pairs), and over 40-fold genome coverage (average 233) (Table 1).

## Genome content analysis

In total, we identified 73 putative RM systems, including 24 type I RM systems, 48 type II RM systems, and 1 type III RM systems [45]. All isolates had at least one type II RM system. GM5, GM16, GM17, OCU468–OCU470, OCU472, OCU473, OCU475, OCU476, OCU479, OCU483 and OCU484 had type I, type II RM systems. GM44 had three types of RM systems. In these RM systems, seven had homologs in MTB and 30 had homologs in *M. kansasii*. Orphan methyltransferase was detected in OCU473 and OCU479. CRISPR was detected only in GM44 (Table 2). The sequences of the region were identical to MAH 104 (query cover: 100%, *E*-value: 0.0, Per. Ident: 99.99%), which is the only MAH strain with an intact CRISPR in the database [51]. The isolates had 185–222 virulence factors; 141 factors were common in all isolates [45]. All isolates shared the same two drug resistance genes: *mtrA*, which is associated with cell division and cell wall integrity [52] and resistance to macrolide antibiotics, and *RbpA,* which regulates bacterial transcription and is associated with rifampicin resistance [45, 53]. In addition, SNPs associated with drug resistance were found. All isolates had a C117D change in the *murA* gene conferring resistance to fosfomycin. An A2274G mutation in the *M. avium* 23S rRNA, which contributes to macrolide resistance, was also detected by RGI, but when we examined the aligned nucleotide sequence, no point mutation was found in any isolates [45]. CRISPR, virulence factor and drug resistance genes

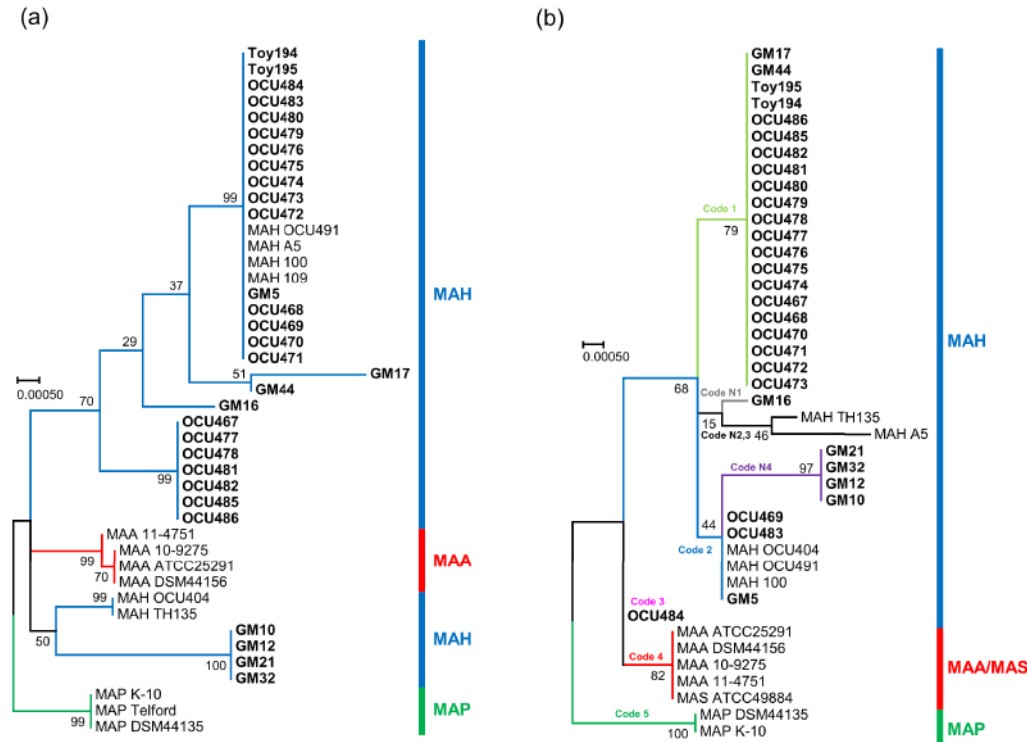

**Figure 3.** Phylogenetic analysis based on *rpoB* gene and *hsp65* gene. Phylogenetic tree was generated with the maximum likelihood method using MEGA 7.0. All isolates in this study are indicated in bold font. (a) 30 MAH isolates in this study were classified as MAH and were differentiated from MAP and MAA nodes. (b) All the isolates in this study were classified into 5 *hsp* code, code 1, 2, 3, N1 and N4. These isolates were differentiated from MAP and MAA/MAS nodes. The bootstrap values were determined from 1,000 replications. The scale bar indicates genetic distances among strains.

were selected from online tools. Original databases of each tool used in this study were updated in 2020, suggesting our data are based on the forefront of existing knowledge.

## tRNA arrays

tRNA arrays were detected in isolates GM17 and OCU479 (Table 3). A tRNA array was discovered in some MAH isolates in a previous study, and phylogenetic analysis based on nucleotide sequences of this tRNA array showed that the tRNA array of MAH was classified into a specific group [40]. Phylogenetic analysis was performed to confirm that the tRNA arrays in this study were authentic. Our tRNA arrays were classified into group 3, as defined in a previous study (Figure 4) [40].

## RE-USE POTENTIAL

MAH is one of the most critical *M. avium* subspecies causing non-tuberculosis mycobacterial infection in human and pigs. Pigs are suspected to be the most dominant host of MAH in animals, and a potential source of infection for humans [7–10]. However, genomic studies on the relationship between human and pig MAH isolates are limited [11, 12]. Our study provides 30 draft genome sequences of MAH isolated from pigs. These data will be useful for genome-based epidemiological studies to evaluate the importance of pigs as a source of infection. In addition, we provide molecular identification of defense

**Table 2.** Detected CRISPR-Cas systems in MAH GM44.

| Strain name | Evidence level | No. of spacers | Sequence of spacers | CRISPR start position | CRISPR end position | CRISPR length | Consensus repeat | Repeat length |
|---|---|---|---|---|---|---|---|---|
| GM44 | 4 | 12 | 1: ACCGGTCGGTCACTGCGGT GGTGTCCTGTGCATGCTCC | 4089 | 4860 | 771 | TGCTCCCCGCGCAA GCGGGGATGAACC | 27 |
| | | | 2: ACCTCCCAGGCGGACGCAGT GCCAGGGATGGCGAGTA | | | | | |
| | | | 3: ACCCGAGGCCGTCGCGGAGG CCTTGACCGACCCCGATA | | | | | |
| | | | 4: ACCGCGCACCTCAGCTGCTG TGCTGCGTGAGCGCGTCATA | | | | | |
| | | | 5: ACCCCTGCACCAGTCGATCCA CTGCGACGTGCGCAGCA | | | | | |
| | | | 6: ACCCATCCCAGGTCAGGAAGT CTGCTCCCCGCGTAAGA | | | | | |
| | | | 7: ACCGGGCCTGTTGCTCATCGG CCCGCCGCGCTCGGGCA | | | | | |
| | | | 8: ACCGCCGATACCGGGCTTGG CATCCGTGCCGTACTGC | | | | | |
| | | | 9: CCCCGTGCCCGGTGGAGGAA CCACCTCTCCCCCCACA | | | | | |
| | | | 10: ACCGGCCGCAGAGGAGGCC GTCACCGCGGCGAAGACC | | | | | |
| | | | 11: ACCCCTCCGATCCAGGTACC GCGTCCGGAAGATGTGGCC | | | | | |
| | | | 12: CCCCCCCCGTCTGCAGCGC AACGGTTCCTACTGCACCTCC | | | | | |

**Table 3.** The information about tRNA array detected in MAH isolates GM17 and OCU479.

| Isolate | Contig | Locus tag | tRNA gene isotype synteny | Species | Query cover | Identity | Accession |
|---|---|---|---|---|---|---|---|
| GM17 | Contig 1 | GBQ13_00450 – GBQ13_00590 | LLKKGCWPVMNYQ QEFEINASHRRLITR | *Mycobacterium chimaera* strain MC045 genome assembly, plasmid: 2 | 64% | 85.97% | LT703506 |
| OCU479 | Contig 38 | GBP94_21805 – GBP94_21975 | TWLLKKGCPVMNY QQEFEIPASHRRLRI | *Mycobacterium chimaera* strain AH16 plasmid unnamed1 | 32% | 76.88% | CP012886 |

systems, tRNA arrays, virulence factors and drug resistance genes. These data are expected to be used in future research on MAH classification, pathogenicity, and identification of antimicrobial drug targets. Principally, our draft genomes were derived from both cases of systemic and lymph node-limited infection of MAH. Thus, the provided virulence factors can be included as important candidate genes associated with the systemic infection of pigs.

## DATA AVAILABILITY

A summary of information about the 30 draft MAH isolate genomes is shown in Table 1. Genome sizes ranged from approximately 4.8 Mbp to 5.6 Mbps. GC content was 68.77–69.26%. All genome sequences have been deposited in GenBank under accession numbers VRUQ00000000, WEGO00000000 to WEGZ00000000 and WEHA00000000 to WEHQ00000000, and in the NCBI Sequence Read Archive (SRA) under accession numbers SRR13521605, SRR13556487 to SRR13556515. Other supporting data underlying the tables, annotations and other results are available in the *GigaScience* GigaDB repository [45].



**Figure 4.** Phylogenetic tree based on the sequence of tRNA isotype located in tRNA array. Phylogenetic tree was generated by maximum likelihood method using MEGA 7.0. Two isolates (GM17 and OCU479 indicated in bold) were classified in Group 3. The bootstrap values were determined from 1,000 replications. The scale bar indicates genetic distances among strains.

## DECLARATIONS
## LIST OF ABBREVIATIONS

BLAST: Basic Local Alignment Search Tool; CRISPR: clustered regularly interspaced short palindromic repeat; MAA: *M. avium* subsp. *avium*; MAC: *Mycobacterium avium* complex; MAH: *Mycobacterium avium* subsp. *hominissuis*; MAP: *M. avium* subsp. *paratuberculosis*; MAS: *M. avium* subsp. *silvaticum*; MEGA: Molecular Evolutionary Genetics Analysis; MTB: *Mycobacterium tuberculosis*; NCBI: National Center for Biotechnology Information; NTM: non-tuberculous mycobacteria; RM: restriction modification; PGAP: Prokaryotic Genome Annotation Pipeline; SNP: single nucleotide polymorphism; SRA: NCBI Sequence Read Archive.

## CONSENT FOR PUBLICATION

Not applicable.

## COMPETING INTERESTS

The authors declare that they have no competing interests.

## FUNDING

This research was supported by a grant from the Japan Agency for Medical Research and Development (AMED) (17fk0108116h040 and 21fk0108129h0502), the Japan Racing Association (JRA) Livestock Industry Promotion Project (H28-29_239, H29-30_7) of the JRA, a grant for Meat and Meat Products (H28-130, H30-60) managed by the Ito Foundation for research in design study, collection, analysis; and was supported by grants from the Japan Society for the Promotion of Science (JSPS) KAKENHI (JP26304039, JP18K19674, 16H05501, 16H01782, 20H00562). JOO is a recipient of a Japanese Ministry of Education, Culture, Sports, Science and Technology (MEXT) scholarship.

## AUTHOR'S CONTRIBUTIONS

T.K., K.O. and H.Y. wrote the manuscript. K.M., A.H., S.S. and K.S. collected samples. K.O., J.O.O., S.S. and K.S. performed laboratory works. T.K., K.O., A.O., H.Y., J.O.O., T.Ito and M.K. conducted computational analysis. Y.N., T.A., T.Y., H.F., T.W., S.Y., K.A. designed methods. M.A., A.D.B., K.O., N.Y., T.Iwamoto and F.M. designed whole research and advised on the interpretation of the study's findings. All authors reviewed the manuscript.

## ACKNOWLEDGEMENTS

We thank the member of Gifu Central Hygiene Service Center and Toyama Meat Inspection Center for sampling. Computational resources were partly provided by the Data Integration and Analysis Facility, National Institute for Basic Biology, Japan.

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
