## [Reviewer Report]

Comments on revised manuscriptMy comments have been sufficiently taken into account and the manuscript is acceptable for publication. 
One additional further comment: 
Line 93: Please correct „MAC has 4 subspecies …” by “M. avium has 4 subspecies…”.

---

## [Reviewer Report]

Reviewer name and names of any other individual's who aided in reviewer Astrid LewinDo you understand and agree to our policy of having open and named reviews, and having your review included with the published papers. (If no, please inform the editor that you cannot review this manuscript.)YesIs the language of sufficient quality?YesPlease add additional comments on language quality to clarify if needed
Are all data available and do they match the descriptions in the paper? YesAdditional CommentsAre the data and metadata consistent with relevant minimum information or reporting standards? See GigaDB checklists for examples <a href="http://gigadb.org/site/guide" target="_blank">http://gigadb.org/site/guide</a>YesAdditional CommentsIs the data acquisition clear, complete and methodologically sound?YesAdditional CommentsIs there sufficient detail in the methods and data-processing steps to allow reproduction?YesAdditional CommentsChapter „Methods, b) Bacterial isolation and DNA extraction”:
Lines 139-140:
There is a discrepancy between the method of DNA extraction as described in the reference (16) and the manuscript text. While according to the reference the bacterial pellet is dissolved in acetone, the manuscript text describes a treatment with chloroform and methanol. This should be clarified.
Is there sufficient data validation and statistical analyses of data quality? Not my area of expertiseAdditional CommentsIs the validation suitable for this type of data?YesAdditional CommentsIs there sufficient information for others to reuse this dataset or integrate it with other data?YesAdditional CommentsAny Additional Overall Comments to the AuthorChapter „Methods, b) Bacterial isolateion and DNA extraction”:
Chapter “Data Validation and quality control, Identification of MAH”
Lines 218-220:
It is true, that the isolates had the highest identity with one of the three MAH strains, but not with all of the three MAH reference strains. For example, isolate OCU468 has 98.69% identity with MAH TH135 but 98.79% identity with MAP K-10. The degree of identity seems to be highly dependent on the choice of strains. Therefore, this comparison may not be very significant. In my experience, growth at 42°C very well distinguishes MAH from the other M. avium subspecies. 
RecommendationAccept

---

## [Reviewer Report]

Reviewer name and names of any other individual's who aided in reviewer Nabeeh HasanDo you understand and agree to our policy of having open and named reviews, and having your review included with the published papers. (If no, please inform the editor that you cannot review this manuscript.)YesIs the language of sufficient quality?YesPlease add additional comments on language quality to clarify if needed
A few minor grammatical edits could be doneAre all data available and do they match the descriptions in the paper? NoAdditional CommentsThe data are not currently accessible by the public on NCBIAre the data and metadata consistent with relevant minimum information or reporting standards? See GigaDB checklists for examples <a href="http://gigadb.org/site/guide" target="_blank">http://gigadb.org/site/guide</a>YesAdditional CommentsIs the data acquisition clear, complete and methodologically sound?YesAdditional CommentsIs there sufficient detail in the methods and data-processing steps to allow reproduction?YesAdditional CommentsIs there sufficient data validation and statistical analyses of data quality? YesAdditional CommentsIs the validation suitable for this type of data?YesAdditional CommentsIs there sufficient information for others to reuse this dataset or integrate it with other data?YesAdditional CommentsAny Additional Overall Comments to the AuthorRecommendationAccept